# Prospective observational study of oxidative stress in the pathology of benign prostatic hyperplasia with bladder diverticulum

**Shuijing Yin, Yu Qiu** [ID]*

Department of Urology, Second Affiliated Hospital of Harbin Medical University, Harbin, Heilongjiang, China

* 821745@hrbmu.edu.cn

## Abstract

### Background

Oxidative stress contributes to benign prostatic hyperplasia (BPH) pathogenesis, but its role in BPH with bladder diverticulum is unclear.

### Methods

This prospective cohort study compared 126 BPH patients at the Second Hospital of Harbin Medical University. The study involved two groups (n = 63 for each group): group A, comprising patients with BPH, and group B, consisting of BPH patients with bladder diverticulum. Ultrasound imaging and CT scans were employed to assess the features of BPH and bladder diverticulum, respectively. Various clinical parameters and oxidative stress biomarkers were compared between the groups.

### Results

Group B exhibited significantly higher creatinine (101.8 ± 27.6 µmol/L vs. 56.1 ± 23.6 µmol/L, p < 0.0001), WBC counts (7.0 ± 1.9 vs. 4.2 ± 1.3 × $10^9$/L, p < 0.0001), residual urine volume (400.1 ± 252.0 mL vs. 150.7 ± 93.9 mL, p < 0.0001), and oxidative stress markers, including 8-OHdG (1.93 ± 0.58 vs. 1.70 ± 0.73 ng/mg creatinine, p = 0.014) and MDA (2.46 ± 0.57 vs. 2.03 ± 0.57 µmol/L, p < 0.0001). In group A, 8-OHdG positively correlated with residual urine volume (rho = 0.68) and nitric oxide with bladder wall thickness (rho = 0.70), while quality of life (QoL) negatively correlated with nitric oxide (rho = -0.76). In group B, oxidative stress markers correlated positively with BMI (e.g., homocysteine, rho = 0.69) and bladder wall thickness (e.g., nitric oxide, rho = 0.69), with QoL negatively correlated with uric acid (rho = -0.78).

**Data availability statement:** All relevant data are within the manuscript and its Supporting Information files.

**Funding:** The author(s) received no specific funding for this work.

**Competing interests:** The authors have declared that no competing interests exist.

## Conclusions

Bladder diverticulum in BPH patients is associated with elevated oxidative stress, increased inflammation, and impaired bladder function.

## Introduction

Benign prostatic hyperplasia (BPH), a common urological condition in elderly males, affects over 50% of those over 50 and up to 90% over 80 due to declining androgen levels[1], causing prostate enlargement and bladder outlet obstruction (BOO) with altered detrusor muscle function [2]. Bladder diverticula, pouch-like extensions of the bladder wall, complicate BPH by potentially causing additional symptoms, including lower and upper urinary tract obstruction when near the ureter, along with risks of stones, tumors, and inflammation [3–5]. Some diverticula can exert pressure on the bladder neck and urethra, causing lower urinary tract obstruction [6]. Additionally, when a diverticulum is located in close proximity to the ureter, it may exert pressure leading to upper urinary tract obstruction, further complicating urinary dynamics. Positioned primarily at the bladder's base and lateral aspects, these diverticula feature weakened walls, susceptible to inflammation-driven cellular infiltration due to infections. Complications, including stone formation, tumor growth, and ureteral compression are also potential outcomes of diverticula [7].

The primary impact of BPH on the detrusor is the induction of BOO [8], which triggers a complex pathophysiological cascade leading to lower urinary tract symptoms (LUTS) [9]. This obstruction causes mechanical stretch stress, altering gene expression and protein synthesis in the detrusor, resulting in changes to its cytoskeleton, contractile proteins, mitochondrial function, and extracellular matrix [10–12]. BOO-induced BPH triggers mechanical stretch stress and subsequent gene expression changes at the messenger ribonucleic acid (mRNA) and micro ribonucleic acid (miRNA) levels. These changes activate key cell signaling pathways, including cytokine and immune response pathways, growth factor signaling (e.g., Transforming Growth Factor Beta [TGF-β], Hepatocyte Growth Factor [HGF], and Insulin-like Growth Factor 1 [IGF-1]), G-protein-coupled receptor (GPCR) pathways (e.g., endothelin and cholecystokinin/gastrin), nitric oxide (NO) signaling, and hypertrophy-related Phosphoinositide 3-Kinase/Protein Kinase B (PI3K/AKT) signaling, mediated by transcription factors such as Activator Protein 1 (AP-1; composed of Jun Proto-Oncogene [JUN] and FBJ Murine Osteosarcoma Viral Oncogene Homolog [FOS]) and Nuclear Factor Kappa-light-chain-enhancer of activated B cells (NFκB) [13].

BPH-induced BOO contributes to LUTS (e.g., frequency, urgency, weak stream), though LUTS can also stem from other conditions, necessitating distinction from BPH-specific obstruction. Clinical predictors like the international prostate symptom score (IPSS) [14], quality of life (QoL) scores, and pertinent biomarkers including prostate-specific antigen (PSA) levels [15], white blood cell count (WBC) [16], creatinine (Cr) [17], and hemoglobin (Hb) [18] are critical, especially in BPH with bladder diverticula. Oxidative stress, an imbalance of reactive oxygen species (ROS) and

antioxidants, drives BPH pathogenesis by promoting DNA damage, angiogenesis via vascular endothelial growth factor (VEGF), and premalignant changes. Oxidative stress, an imbalance ofROS and antioxidants, drives BPH pathogenesis by promoting DNA damage, angiogenesis via vascular endothelial growth factor (VEGF), and premalignant changes [19–23].

Key oxidative stress biomarkers—uric acid [24], nitric oxide [25], MDA [19], 8-OHdG [26], and homocysteine [27]—are linked to ROS and BPH risk but are underexplored in BPH patients with bladder diverticula. This prospective cohort study investigates oxidative stress in BPH, hypothesizing a stronger association in patients with bladder diverticula, aiming to uncover mechanistic differences and clinical implications by comparing these groups.

## Materials and methods

### Participants

Written informed consent was obtained from all participants prior to their inclusion in the study, ensuring they understood the study's purpose, procedures, potential risks, and benefits. For each participant, the consent process was documented in their medical records, and a copy of the signed consent form was securely stored in compliance with institutional policies. Since this study did not involve minors, no parental or guardian consent was necessary. The ethics committee approved the informed consent procedures as part of the study's ethical review (Approval No.: KY2022–284). Patients diagnosed with BPH between November 15th, 2022 and May 15th, 2023 were recruited from the hospital's urology department. The study starts on November 15th, 2022 and ends on December 15th, 2023.

In our study, BPH was diagnosed based on clinical and ultrasound examinations. Diagnostic criteria included evidence of prostate enlargement with a prostate volume greater than 10 cc, as measured by transrectal ultrasound, in conjunction with an IPSS greater than 1, indicating the presence of significant urinary symptoms. LUTS were assessed independently using the IPSS questionnaire, which evaluates both obstructive and irritative symptoms. This tool categorizes symptoms as mild (IPSS score 0–7), moderate (8–19), or severe (20–35). The presence of LUTS was considered significant for inclusion if the symptoms persisted for more than six months, ensuring chronic symptoms. To differentiate the effects of BPH from other causes of LUTS, patients underwent a thorough medical evaluation to exclude other potential causes such as urinary tract infections, bladder stones, and neurogenic bladder. This approach allowed us to ascertain that the observed LUTS were predominantly due to BPH in the enrolled subjects.

### Prospective cohort study design

The prospective cohort study evaluates two distinct cohorts of patients diagnosed with BPH. A total of 126 patients, who sought medical attention at our institution were evenly divided into two groups: group A, consisting of patients solely afflicted with BPH, and group B, comprising patients with both BPH and bladder diverticulum. A computer-generated random allocation sequence was used to ensure unbiased assignment of participants to each group, maintaining an equal allocation ratio of 1:1 between the two cohorts, with assignments concealed in sequentially numbered, opaque, sealed envelopes. These envelopes were opened only after the enrolled participants completed all baseline assessments, ensuring allocation concealment until prospective cohort study were assigned. This design allows for a straightforward comparison of outcomes between the two groups under study. The data for this study were collected at the Second Hospital of Harbin Medical University, a renowned medical institution known for its comprehensive health-care services and advanced research facilities. The hospital, located in Harbin, China, provided a conducive environment for the study, ensuring access to a diverse patient population and state-of-the-art diagnostic and therapeutic resources. The clinical settings within the hospital facilitated thorough and consistent data collection, contributing to the reliability and validity of the study outcomes. BPH was confirmed via ultrasound and bladder diverticulum was primarily diagnosed through CT.

## Sample size calculation

To calculate the sample size for a study with two groups, aiming for a power of 0.80 (80%) and an alpha of 0.05 (5%), with an anticipated effect size (standardized difference between means) of 0.5 standard deviations, we use a formula for sample size calculation in a two-sample t-test scenario. The required sample size per group is approximately 62.8, and thus at least 63 participants in each group.

## Including criteria

Eligible participants were those diagnosed with BPH, confirmed through clinical examination and diagnostic tests such as digital rectal examination and PSA levels. Additionally, for group B, patients with documented bladder diverticulum identified via ultrasonography or CT were included.

## Excluding criteria

Patients were excluded from the study if they had a history of prostate or bladder surgery. Additionally, individuals with a history of urinary tract cancers, severe urinary tract infections, or other major urological disorders were also excluded to maintain the study's focus on BPH and bladder diverticulum. Men with neurological disorders affecting bladder function were excluded. Furthermore, individuals diagnosed with any form of urinary tract cancer were not eligible for inclusion, as these conditions could significantly influence urinary tract function and potentially overlap with the study's diagnostic criteria. Severe urinary tract infections and other major urological disorders were also grounds for exclusion, due to their capacity to alter urinary function and symptoms in a manner similar to the conditions under investigation. Lastly, men suffering from neurological disorders that affect bladder function were excluded, as these could introduce additional variability into the assessment of bladder health and function, thereby compromising the study's ability to accurately isolate the effects of BPH and bladder diverticulum.

## Primary outcomes

The IPSS was systematically evaluated through a questionnaire for all patients. This assessment tool consists of seven items that quantify the severity of LUTS. This score was used to categorize the severity of the symptoms as mild (0–7), moderate (8–19), or severe (20–35)[28]. The incontinence impact questionnaire-7 (IIQ-7) was used to evaluate the impact of urinary incontinence on QoL. The FPSA and TPSA were quantified using ELISA kits (ALPCO, Salem, NH, USA). For analyzing LEU levels and WBC count, the analysis was conducted using automated hematology analyzers (Beckman Coulter, Brea, CA, USA). Hb was measured using the cyanmethemoglobin method.

## Bladder dysfunction assessment

Cystometric measurements were obtained from all patients using a multichannel urodynamic system (Laborie Medical Technologies, Mississauga, Canada), which is equipped to assess urethral pressure profile (Qura), vesical pressure (Pves), and abdominal pressure (Pabd). In the presence of bladder diverticula, detrusor pressure (Pdet) was measured during the voiding phase, which is crucial for evaluating the actual detrusor muscle function. During voiding, the detrusor pressure reflects the effort required to expel urine. For patients with bladder diverticula, measuring Pdet during the voiding phase can provide insight into how these structures affect the overall bladder pressure and emptying efficiency. To mitigate the impact of bladder diverticula on pDet measurements, we have explored alternative analytical methods that could offer more reliable insights. One approach is the segmentation of bladder regions via imaging techniques to isolate the diverticula from the main bladder body in our analysis. By applying pressure measurements only to regions without diverticula, we aim to obtain a more accurate representation of detrusor muscle function. Furthermore, recognizing the complexity introduced by diverticula, we have also incorporated additional metrics such as bladder wall thickness and urinary

flow rates to provide a more comprehensive evaluation of bladder outlet obstruction and detrusor strength. These adjustments allow for a more nuanced understanding of the urinary dynamics in BPH patients, especially those complicated by the presence of bladder diverticula.

Detrusor overactivity (DO) is characterized by involuntary detrusor contractions during the filling phase, which can be identified through elevated Pdet values. Factors influencing DO include low bladder compliance and pathological detrusor responses to filling. Vesicoureteral reflux (VUR) involves the backward flow of urine from the bladder into the ureters or kidneys. High bladder pressures during voiding or due to DO during filling can exacerbate or contribute to VUR, highlighting the importance of understanding Pdet values in the context of bladder and ureteral health.

For other bladder complications, Intravesical protrusion (cm²) was primarily assessed during the filling phase. The intravesical protrusion can be related to bladder outlet obstruction or other structural abnormalities, indicating potential complications such as BPH in men. Residual urine volume (mL) was measured immediately after the voiding phase, as it quantifies the amount of urine left in the bladder after a voiding attempt. A significant residual urine volume can indicate voiding dysfunction or bladder outlet obstruction, leading to complications such as urinary tract infections. Bladder wall thickness (mm) was assessed during both phases but was often measured during the filling phase to evaluate the bladder in a more static state. Increased bladder wall thickness is associated with bladder outlet obstruction and detrusor overactivity, reflecting chronic bladder conditions such as overactive bladder syndrome and chronic urinary retention. Max urinary flow rate (mL/s) was measured during the voiding phase. The maximum urinary flow rate is a critical parameter in diagnosing obstructive and some neurogenic bladder disorders. Lower rates can indicate bladder outlet obstruction or detrusor muscle weakness. Detrusor pressure at Qmax (cm $H_2O$) was measured during voiding phase. This measures the pressure exerted by the bladder muscle (detrusor) at the maximum flow rate, helping to identify issues like bladder outlet obstruction or detrusor underactivity/overactivity.

The assessment of bladder dysfunction through cystometric measurements is critically influenced by both environmental conditions and the patient's emotional state [29]. To optimize the environment for cystometric measurements and mitigate the influence of external factors, specific environmental adjustments are essential. The examination room should maintain a comfortable temperature between 20°C to 22°C (68°F to 71.6°F), with noise levels not exceeding 30 dB, to create a tranquil and stress-minimizing setting. The lighting should be kept at soft, natural levels, ideally between 200–500 lux, to avoid the clinical ambience often associated with medical settings. Additionally, the equipment used for the measurements should be calibrated regularly, considering an atmospheric pressure of 1013 hPa and a humidity level maintained at 45–55%, ensuring the accuracy and reliability of the readings. These environmental controls are critical in establishing a stable setting that minimizes physiological stress responses potentially affecting the outcome of the procedure.

Addressing patient emotional distress involves a comprehensive approach that incorporates pre-procedure preparation and in-procedure support mechanisms. Providing patients with detailed verbal and written explanations about the procedure can significantly alleviate anxiety, with studies suggesting a reduction in patient stress by up to 40%. Ensuring complete privacy during the examination, through the effective use of partitions or curtains, enhances patient comfort. The introduction of guided relaxation techniques for 5–10 minutes before the procedure has been shown to effectively reduce indicators of stress, such as heart rate and blood pressure. Furthermore, allowing the presence of a supportive healthcare professional or a patient-chosen companion during the examination can further decrease anxiety levels by up to 30%.

## Measurement of oxidative stress biomarkers

Various biomarkers were quantified using kits supplied by Sangon Biotech (Shanghai) Co., Ltd., China. The 8-OHdG levels in urine were measured via an ELISA kit. MDA was assessed using the TBARS assay; Nitric oxide metabolites were determined colorimetrically after enzymatic reduction and reaction with Griess Reagent. Homocysteine concentrations were evaluated using a chemiluminescent immunoassay based on competition binding and measured with a luminometer. Uric acid was quantified enzymatically, with the resultant color intensity, indicative of concentration, read at 520 nm.

## Secondary outcomes

The correlation analysis assessed relationships between clinical parameters and oxidative stress biomarkers in the group A and the group B using Pearson correlation coefficients. The parameters included IPSS, QoL, BMI, FPSA, TPSA, WBC count, Hb, creatinine levels, intravesical protrusion, residual urine volume, and maximum urinary flow rate. Oxidative stress biomarkers analyzed were 8-OHdG, MDA, nitric oxide, homocysteine, and uric acid. The diverticulum volume was measured using CT imaging and included in the analysis. Statistically significant correlations ($p < 0.05$) and 95% confidence intervals were utilized to ensure the robustness of the findings. The results revealed significant correlations between oxidative stress markers and clinical parameters, demonstrating distinct patterns between the two groups.

For each primary outcome (oxidative stress biomarkers: 8-OHdG, MDA, nitric oxide, homocysteine, uric acid) and secondary outcomes (clinical parameters: IPSS, QoL, BMI, FPSA, TPSA, WBC count, Hb, creatinine levels, intravesical protrusion, residual urine volume, maximum urinary flow rate), group B displayed significantly higher levels and more severe clinical presentations than group A, with estimated effect sizes for key differences (e.g., creatinine levels, WBC count, residual urine volume) showing 95% confidence intervals indicating robust statistical significance ($p < 0.05$). No serious adverse events were reported in either group, with zero adverse events reported. Both groups experienced only minor side effects, such as mild discomfort during ultrasound and CT procedures, which were transient and resolved.

## Statistical analysis

Intergroup comparisons of variables were conducted via the student's t-test. Count data were analyzed via the Chi-Square test. Statistical significance was deemed present when $p < 0.05$. To enhance the robustness of the findings, 95% confidence intervals were used to account for potential biases and ascertain more reliable conclusions.

## Results

### Baseline clinical characteristics

In an elderly male cohort, no significant age discrepancy was found between group A ($74.1 \pm 13.9$ years) and group B ($72.2 \pm 11.9$ years, $p = 0.359$, n = 63 for each group, Table 1), nor were differences in IPSS, QoL, BMI, hemoglobin levels, or intravesical protrusion ($p > 0.05$) observed, suggesting a similar age profile and baseline clinical characteristics among BPH patients, with or without bladder diverticulum. Smoking has been linked to inflammation and changes in hormone levels that could exacerbate BPH symptoms or influence the disease's progression [30]. Smoking status alone did not differentiate the clinical presentations or outcomes of BPH between patients with and without bladder diverticulum ($p = 0.543$, Table 1).

### The features of BPH and bladder diverticulum

This flow diagram in Fig 1 illustrates the prospective cohort study design, starting with the recruitment phase where patients with BPH seeking medical attention are identified. The process continues with a baseline assessment. The patients are then allocated into two groups (n = 63 for each group): group A, consisting of patients solely with BPH, and group B, consisting of patients with both BPH and bladder diverticulum. Ultrasonographic and computed tomography imaging provide distinct insights into BPH with or without bladder diverticula. Group A ultrasound images exhibit the prostate's typical BPH characteristics: a length of 2.48 cm, width of 2.45 cm and a smoothly contoured bladder wall devoid of diverticula (Fig 2A). Similarly, a length of 10.57 cm, width of 4.84 cm, height of 11.46 cm, and a homogeneous echotexture without focal lesions (Fig 2B). and a homogeneous echotexture — Contrastingly, CT scans of group A illustrate a smoothly contoured bladder wall length of 11.5 cm and width of 5.23 cm corroborate BPH's diagnostic enlargement, with a hypoechoic presentation and uniform gland texture (Fig 2C). Group B's CT imagery reveals bladder diverticula as dark, urine-filled outpouchings adjacent to the bladder, devoid of surrounding tissue density irregularities, indicative of BPH with bladder diverticulum (Fig 2D).

**Table 1. Basic information of both groups.**

| Parameters | group A (n=63) | group B (n=63) | p-value |
|---|---|---|---|
| Age | 74.1±13.9 | 72.2±11.9 | 0.359 |
| IPSS | 14.9±6.2 | 14.6±7.8 | 0.835 |
| Mild, n (%) | 0 (0%) | 0 (0%) | |
| Moderate, n (%) | 10 (15.9%) | 13 (20.6%) | 0.645 |
| Severe, n (%) | 53 (84.1%) | 50 (79.4%) | |
| Smoking, n (%) | 15 (23.8%) | 18 (28.6%) | 0.543 |
| QoL | 9.8±5.2 | 9.3±4.6 | 0.568 |
| BMI (kg/m²) | 22.9±2.2 | 22.7±2.4 | 0.771 |
| FPSA (ng/mL) | 3.7±2.2 | 2.8±1.8 | 0.015 |
| TPSA (ng/mL) | 11.1±7.6 | 11.9±5.7 | 0.465 |
| FPSA/TPSA% | 37.3±12.2 | 28.2±17.7 | 0.001 |
| Creatinine (μmol/L) | 56.1±23.6 | 101.8±27.6 | <0.0001 |
| WBC ($10^9$/L) | 4.2±1.3 | 7.0±1.9 | <0.0001 |
| Hb (g/L) | 131.4±9.3 | 134.4±12.2 | 0.115 |
| Alpha-blockers, n (%) | 40 (63.5%) | 35 (55.6%) | 0.364 |
| 5-alpha-reductase inhibitors, n (%) | 20 (31.7%) | 25 (39.7%) | 0.353 |
| **Diverticulum Volume (cc)** | – | 12.45±4.32 | |
| Prostate Volume (cc) | 45.0±31.6 | 47.6±33.4 | 0.658 |
| Intravesical Protrusion (cm²) (filling phase) | 6.9±3.4 | 6.5±3.9 | 0.550 |
| Residual Urine Volume (ml) (post voiding) | 150.7±93.9 | 400.1±252.0 | p<0.0001 |
| Kidney stones (cm in diameter) | 0.26±0.09 | 0.30±0.11 | 0.04 |
| Bladder wall thickness (mm) (voiding and filling phase) | 3.93±0.73 | 4.76±0.88 | p<0.0001 |
| Max urinary flow rate (mL/s) (voiding phase) | 14.4±3.39 | 13.34±3.37 | 0.073 |
| Detrusor pressure at Qmax (cm $H_2O$) (voiding phase) | 114.7±38.4 | 119.1±35.8 | 0.51 |
| Detrusor overactivity (cm $H_2O$) (filling phase) | 31.2±11.7 | 33.6±12.8 | 0.085 |
| Vesicoureteral reflux (VUR) (filling and voiding phases) | | | |
| Grade I (%) | 15(23.8) | 12(19) | 0.881 |
| Grade II (%) | 24(38.1) | 23(36.5) | |
| Grade III (%) | 19(30.2) | 22(34.9) | |
| Grade IV (%) | 5(7.9) | 6(9.5) | |
| Other diseases | | | |
| Hypertension, n (%) | 5 (7.9%) | 6 (9.5%) | 0.794 |
| Diabetes, n (%) | 6 (9.5%) | 4 (6.3%) | |
| Hyperlipidemia, n (%) | 6 (9.5%) | 6 (9.5%) | |

Note: International Prostate Symptom Score; QoL: Quality of Life; BMI (kg/m²): Body Mass Index (kilograms per square meter); FPSA: Free Prostate-Specific Antigen; TPSA: Total Prostate-Specific Antigen. WBC ($10^9$/L): White Blood Cell count. Hb (g/L): Hemoglobin. P-values less than 0.05 are generally considered to indicate statistically significant differences between groups. The results show statistically significant differences in several parameters, including IPSS, TPSA, creatinine levels, WBC count, intravesical protrusion, residual urine volume, and maximum urinary flow rate, suggesting distinct clinical presentations between the two groups, BPH patients (group A) and BPH patients with bladder diverticulum (group B). The diverticulum volume was measured using computed tomography (CT) imaging, and the results are presented as mean±standard deviation (SD).

The CT scans presented in Figs 2E and 2F from group A, which consists of patients with BPH only, show no evidence of bladder diverticulum, confirming the absence of any urine-filled outpouchings from the bladder wall. In contrast, Figs 2G and 2H from group B, consisting of patients with both BPH and bladder diverticulum, clearly display the presence of bladder diverticula. These diverticula are identifiable as dark, urine-filled outpouchings extending from the bladder wall. The distinction between the two groups is evident, with group A exhibiting normal bladder morphology and group B showing

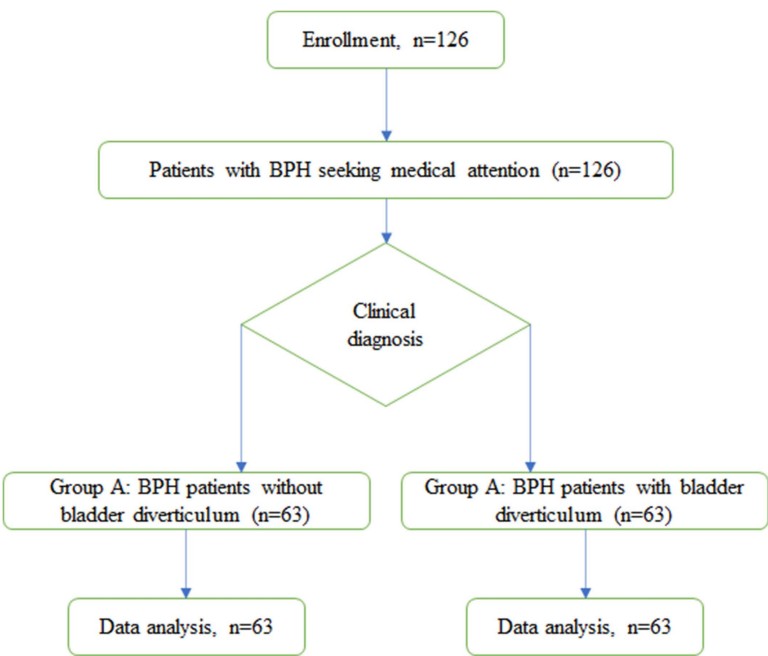

**Fig 1. The flow diagram of this study.** The patients with BPH seeking medical attention are identified. The process continues with a baseline assessment. The patients are then allocated into two groups: group A, consisting of patients solely with BPH, and group B, consisting of patients with both BPH and bladder diverticulum.

pathological changes consistent with bladder diverticulum. This visual difference underscores the significance of bladder diverticulum in the pathology of patients with BPH.

## Primary outcomes

Primary outcomes of all 126 patients from the two groups can be available. Alpha-blockers[31] and 5-alpha-reductase inhibitors [32] play pivotal roles in BPH management by improving urinary flow and reducing prostate size, respectively. These medications could significantly affect outcomes like Max Urinary Flow Rate, Residual Urine Volume, Prostate Size, and Intravesical Protrusion. Alpha-blockers and 5-alpha-reductase inhibitors did not differentiate the clinical presentations or outcomes of BPH between the two groups either (p = 0.364 and 0.353 respectively). Conversely, prostate-specific and urinary metrics did vary significantly; group B had lower FPSA levels (2.8 ± 1.8 vs. 3.7 ± 2.2, p = 0.015) and fPSA/tPSA% (28.2 ± 17.7 vs. 37.3 ± 12.2, p = 0.001), hinting at divergent underlying pathologies. Elevated creatinine (101.8 ± 27.6 μmol/L vs. 56.1 ± 23.6 μmol/L, p < 0.001), WBC counts (($7.0 ± 1.9$) ×$10^9$/L vs. ($4.2 ± 1.3$) ×$10^9$/L, p < 0.0001) and residual urine volume (400.1 ± 252.0 mL vs. 150.7 ± 93.9 mL, p < 0.0001) in group B implied renal impairment and increased inflammatory or infectious processes due to bladder diverticulum. Despite similar prostate sizes (p = 0.658), group B exhibited significantly greater residual urine, bladder wall thickness, emphasizing the profound effects of bladder diverticulum on the urinary system in BPH patients. In the group B, the mean volume of the diverticulum was 12.45 ± 4.32 cc (Table 1). This parameter was analyzed to assess its potential impact on oxidative stress biomarkers and clinical outcomes.

## Bladder pressure dynamics in BPH patients with and without bladder diverticulum

In a comparative study assessing bladder pressure dynamics between group A and group B, both groups demonstrated an ascending trend in Pdet values, yet they differed significantly in the variability and extremity of these values (Figs 3A

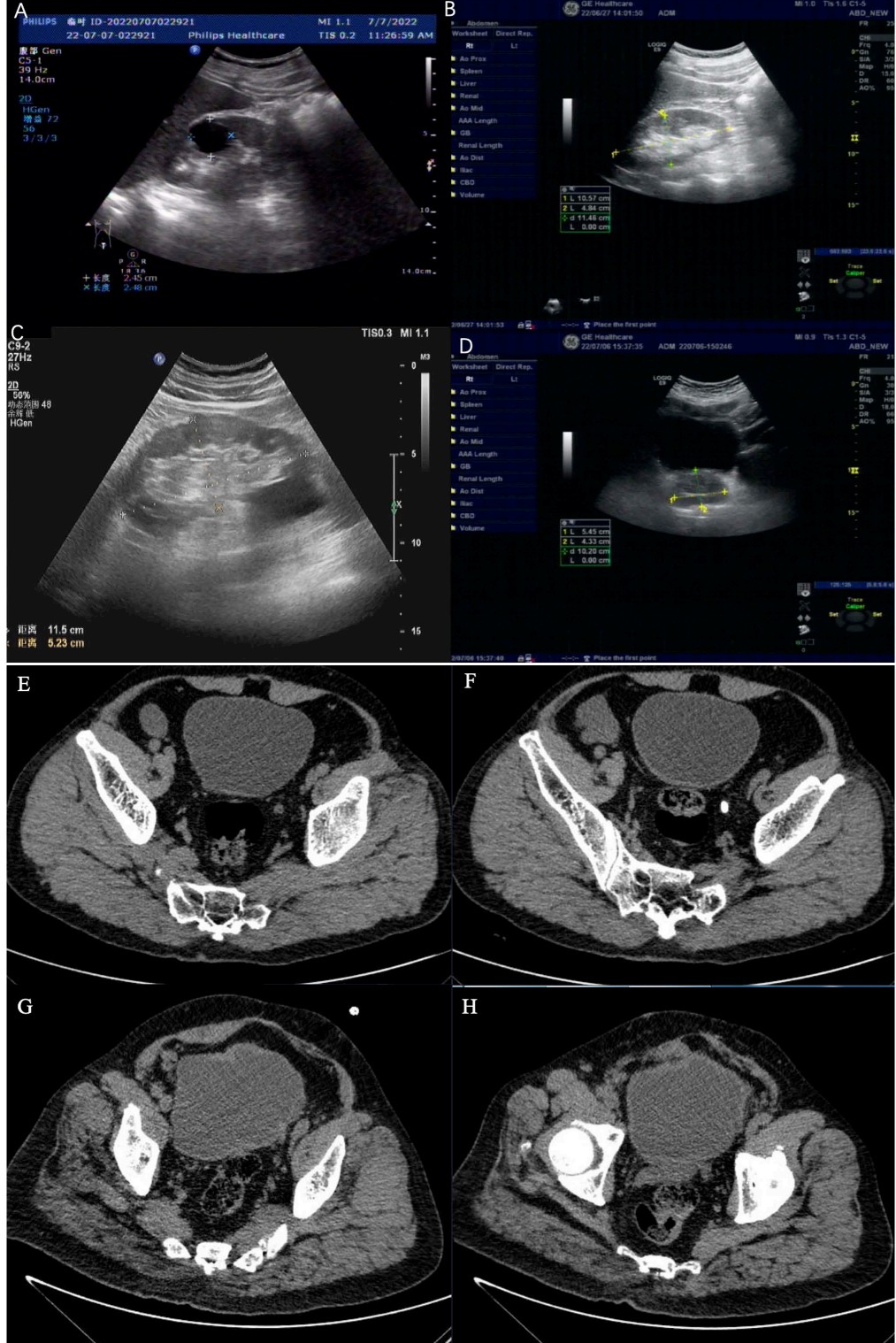

**Fig 2. Ultrasonographic and computed tomographic features of benign prostatic hyperplasia and bladder diverticulum.** Figs 2A–2D showcase ultrasonographic evidence of benign prostatic hyperplasia (BPH) with characteristic gland enlargement and homogeneity without nodules or cysts. Figs

A and B from group A with BPH without bladder diverticulum. Figs 2C and 2D from group B with BPH and bladder diverticulum. CT scans from group A (Figs 2E and 2F) without bladder diverticulum, exhibit the absence of bladder diverticulum, while Figs 2G and 2H from group B reveal the presence of diverticula as dark, urine-filled outpouchings from the bladder wall.

and 3B). group A showed a steady increase in Pdet, with a maximum range of 65.2 to 180.4 cm $H_2O$ and a minimum range of -20.2 to -4.6 cm $H_2O$, suggesting uniform progression in bladder pressure dynamics. Conversely, group B exhibited a broader and higher range of Pdet values, from 59.4 to 206.3 cm $H_2O$, indicating more variable and severe detrusor pressure peaks. This trend was also reflected in Pves measurements, where group A displayed significant variability with maximum Pves reaching 179.8 cm $H_2O$, and group B showed even higher peaks, with the third subgroup in a wider range from -20.6 cm $H_2O$ to 194.6 cm $H_2O$. Despite these differences in Pdet and Pves, parameters such as urine flow rate (Qura), electromyography (EMG), and abdominal pressure (Pabd) remained consistently stable across both groups, indicating that the presence of bladder diverticulum did not markedly affect these aspects of urinary tract functionality. These findings underscore the nuanced clinical management required for BPH patients with diverticulum, given their more complex bladder dynamics.

### Diverticulum-BPH oxidative biomarker patterns compared to BPH alone

The comparative analysis of oxidative stress biomarkers between group A and group B revealed distinct disparities. Group B exhibited a significantly higher mean level of 8-OHdG (1.93±0.58 ng/mg creatinine), compared to group A (1.70±0.73 ng/mg creatinine), indicating a more severe oxidative impact on DNA (Table 2, p=0.014). Similarly, markers of lipid peroxidation and nitric oxide metabolism, namely MDA (2.46±0.57 µmol/L vs. 2.03±0.57 µmol/L, p=0.0007) and nitric oxide metabolites (298.8±106.9 vs. 217.4±119.1 µmol/L creatinine, p=0.0001), were significantly elevated in group B. This trend continued with higher levels of homocysteine (13.74±3.80 µmol/L vs. 9.57±3.13 µmol/L, p=0.0001) and uric acid (5.68±1.48 mg/dL vs. 5.10±1.51 mg/dL, p=0.0157) in group B (Table 2). These statistically significant findings revealed a heightened oxidative burden in group B.

### Secondary outcomes: correlation matrix of clinical and oxidative stress biomarkers in group A

The study confirms the pivotal role of oxidative stress biomarkers in the pathology of BPH absent bladder diverticulum, as depicted in a comprehensive correlation matrix (Fig 4). The strongest positive correlations involving oxidative stress factors indicate significant correlations between specific health parameters. 8-OHdG shows the strongest positive correlation with residual urine volume (0.68) and WBC (0.67), underscoring its potential role in these conditions. Similarly, Nitric Oxide exhibits strong positive correlations with bladder wall thickness (0.70) and BMI (0.68), suggesting its significant influence on these parameters. These relationships highlight the multifaceted role of oxidative stress in the progression of BPH.

Conversely, the top negative relationships reveal areas where oxidative stress might detrimentally impact health parameters. QoL has the strongest negative correlation with Nitric Oxide (-0.76), indicating that higher levels of this oxidative stress marker are associated with poorer QoL. Additionally, Creatinine (-0.70), MDA (-0.55), and Uric Acid (-0.52) also show negative correlations with QoL, emphasizing the adverse effects of elevated oxidative stress on overall well-being. These findings underscore the importance of managing oxidative stress to maintain overall health and quality of life.

### Secondary outcomes: correlation matrix of clinical and oxidative stress biomarkers in group B

In patients with BPH accompanied by bladder diverticulum, the correlation matrix analysis (Fig 5) elucidates a substantial link between oxidative stress biomarkers and BMI. 8-OHdG (0.62), MDA (0.62), Nitric Oxide (0.68), and Homocysteine (0.69) all exhibit significant positive correlations with BMI, indicating a close relationship between oxidative stress and

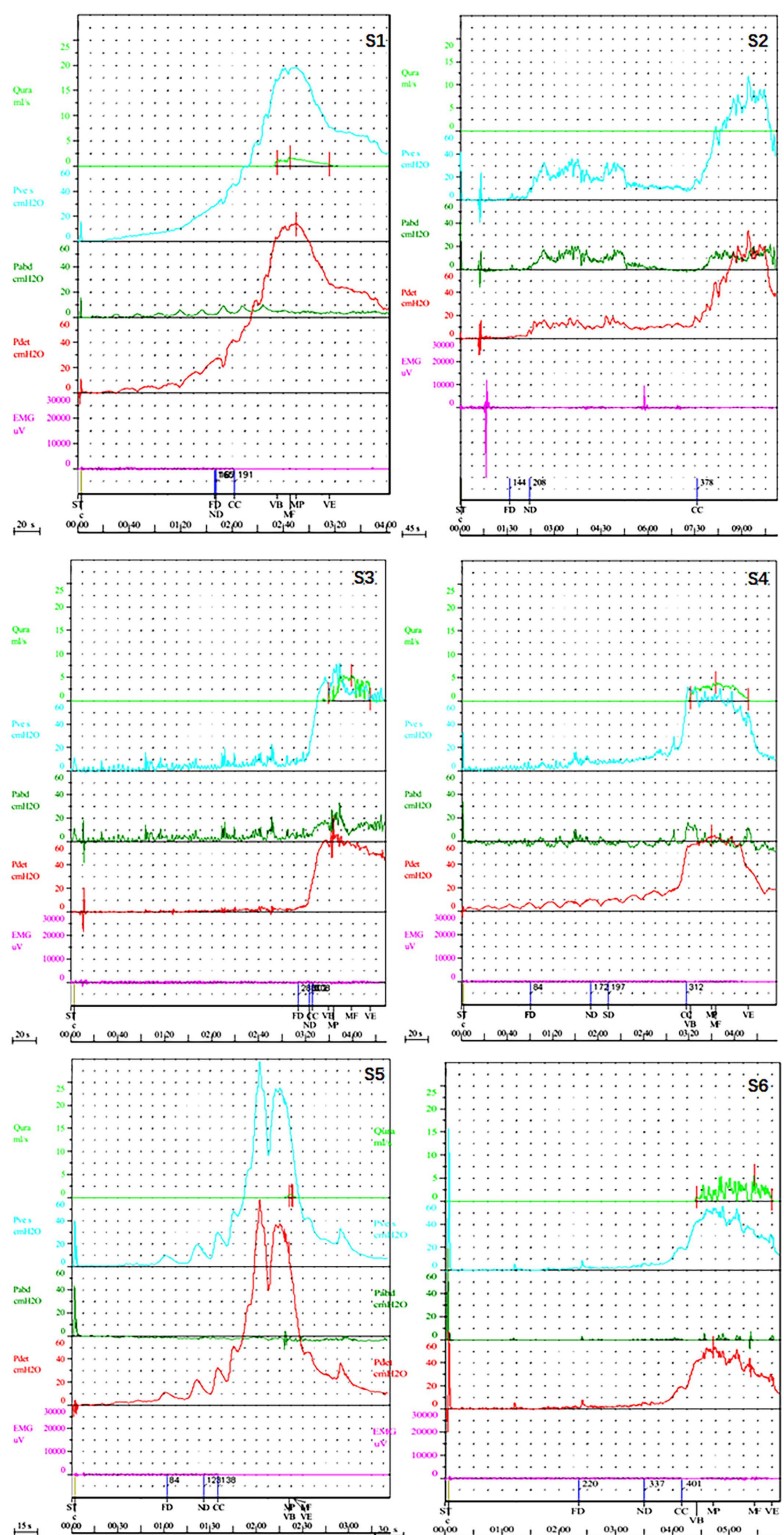

**Fig 3. Impact of bladder diverticulum on the detrusor pressure dynamics in BPH patients.** Fig 3A depicts vesical pressure (Pves) and detrusor pressure (Pdet) trends in benign prostatic hyperplasia (BPH) patients without bladder diverticulum (group A). Fig 3B illustrates the Pves and Pdet trends in BPH patients with bladder diverticulum (group B), where Pves and Pdet values also increase but with a wider range.

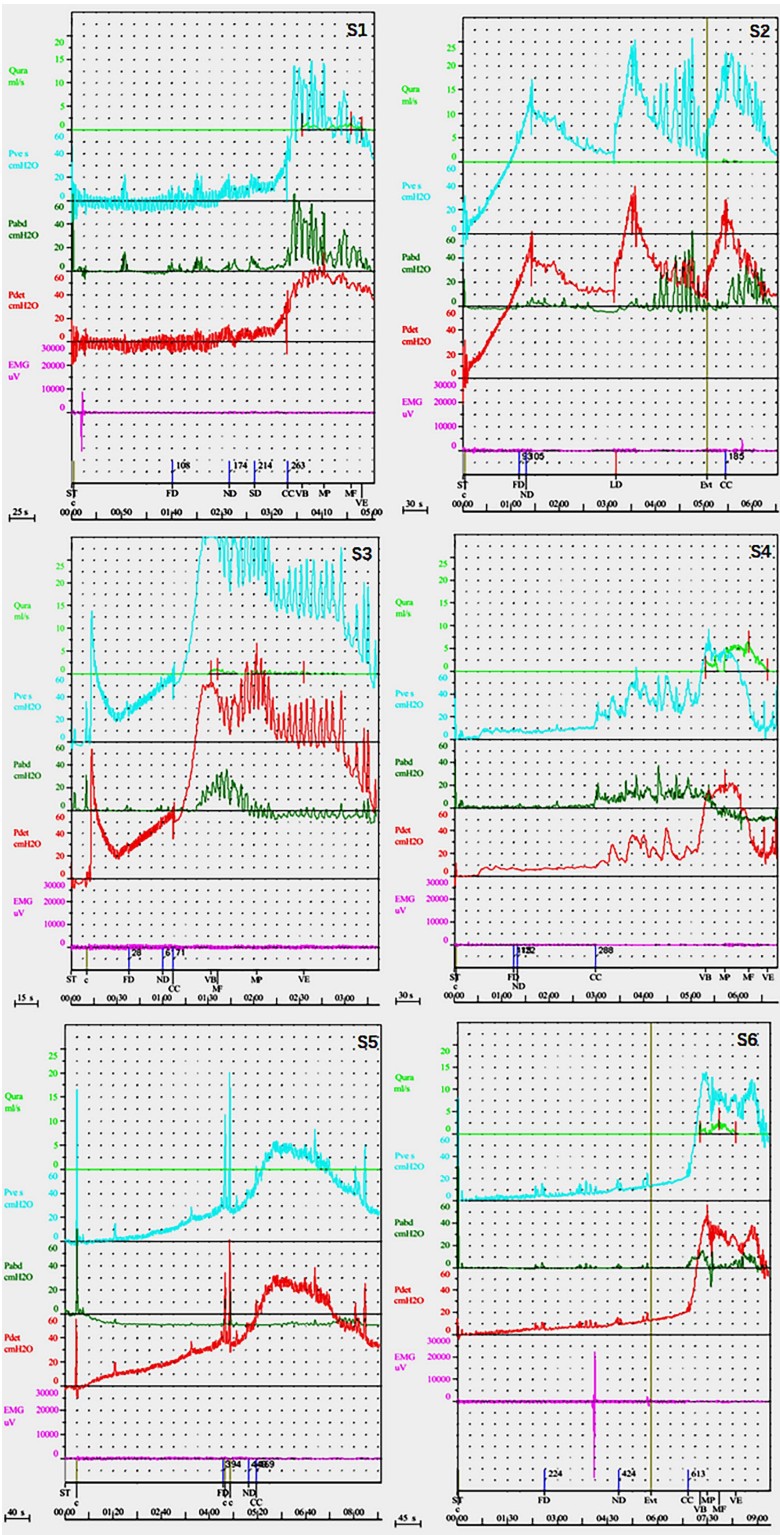

**Fig 3.** Continued.

**Table 2. Oxidative biomarker levels between two groups (Mean ± SD).**

| Biomarker | group A | group B | p-value |
|---|---|---|---|
| 8-OHdG (ng/mg creatinine) | 1.70 ± 0.73 | 1.93 ± 0.58 | 0.0142 |
| MDA (μmol/L) | 2.03 ± 0.57 | 2.46 ± 0.57 | <0.0001 |
| Nitric oxide | 217.4 ± 119.1 | 298.8 ± 106.9 | <0.0001 |
| Homocysteine (μmol/L) | 9.57 ± 3.13 | 13.74 ± 3.80 | <0.0001 |
| Uric Acid (mg/24h) | 5.10 ± 1.51 | 5.68 ± 1.48 | 0.0157 |

Note: n = 63 for each group. The statistical differences are significant if p < 0.05.

body mass in these patients. Additionally, Uric Acid stands out with a strong positive correlation with WBC (0.68), suggesting an association with immune response.

Similarly, Nitric Oxide shows a significant positive association with bladder wall thickness (0.69), which may reflect inflammatory and vascular changes pertinent to the clinical features of BPH. The pronounced correlations of bladder wall thickness with 8-OHdG (0.63) and Homocysteine (0.64) further indicate the complex interplay of oxidative stress, inflammation, and bladder tissue remodeling in the disease. The negative relationships between QoL and oxidative stress biomarkers, specifically 8-OHdG (-0.56), MDA (-0.29), Nitric Oxide (-0.60), and Uric Acid (-0.78), underscore the detrimental impact of oxidative stress on patients' quality of life. These correlations highlight the necessity for targeted therapeutic strategies to mitigate oxidative stress and improve clinical outcomes in BPH patients.

## Discussion

This comparative study distinctly highlights the pathophysiological and clinical differences between group A and group B. Significant differences were observed in WBC, creatinine, residual urine volume and bladder wall thickness (Table 1). Despite these differences in symptom severity, QoL scores are similar across both groups, suggesting that the perceived impact on daily living may not directly correlate with the severity of symptoms. The results are consistent with the previous report that IPSS itself is not an independent predictor of QoL [33]. group B's significantly higher WBC suggests a possible link between bladder diverticulum and urinary tract inflammation, reinforcing the notion that diverticula may foster an environment prone to immune responses and infections [34].

The findings of this study offer a novel perspective on the role of oxidative stress in the pathology of BPH, particularly in patients with bladder diverticulum. The elevated levels of oxidative stress biomarkers, such as uric acid (rho = 0.65) and nitric oxide (rho = 0.69), in group B (BPH with bladder diverticulum) underscore a potential pathophysiological mechanism driving the disease's progression in this subgroup with bladder diverticulum. Uric acid can be associated with bladder obstruction. It mentions that urine sedimentation in the bladder, which can lead to urinary obstruction and pain, can be composed of different materials, including uric acid [35]. Oxidative stress, characterized by an imbalance between the production of reactive oxygen species and the body's ability to detoxify their harmful effects, has been increasingly recognized as a key factor in various urological conditions. In the context of BPH with bladder diverticulum, the strong positive correlations between oxidative stress markers and bladder wall thickness suggest that oxidative stress is associated with bladder wall remodeling and the overall severity of BPH symptoms. This relationship implies that oxidative stress not only plays a role in the etiology of BPH but also influences its progression. These observations highlight the complexity of the interplay between oxidative stress and BPH, especially in patients with bladder diverticulum. The results suggest that oxidative stress biomarkers could serve as potential indicators of disease severity and progression in this patient subgroup. Furthermore, these biomarkers may offer targets for therapeutic strategies aimed at mitigating oxidative damage and managing BPH symptoms. Therapeutic interventions that modulate oxidative stress could thus represent a novel approach to treating BPH, particularly in cases complicated by bladder diverticulum.

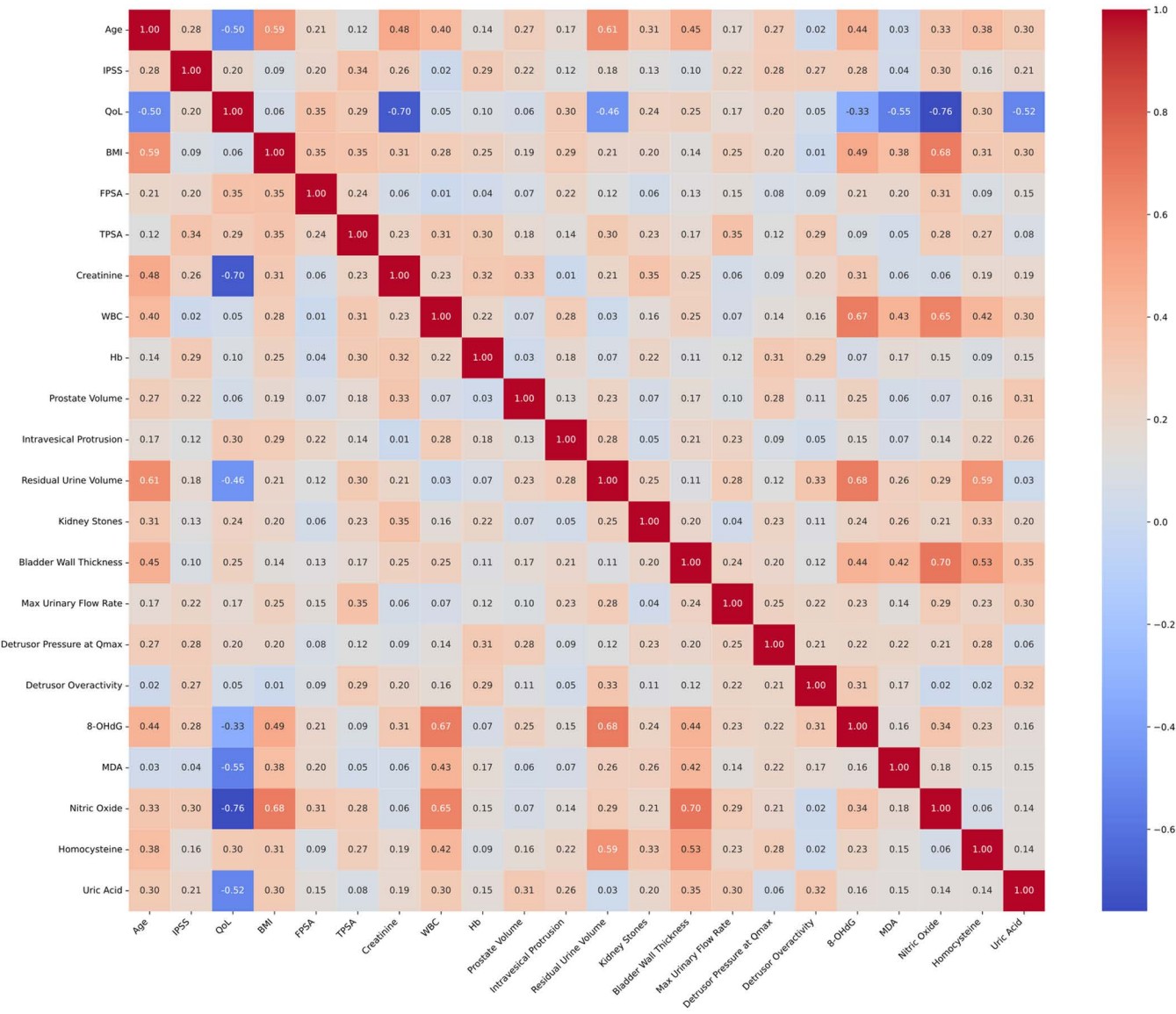

**Fig 4. Correlation matrix of clinical and oxidative stress biomarkers in BPH patients without bladder diverticulum.** The figure illustrates a correlation matrix heatmap with overlaid colored ellipses, representing the relationships between various clinical and biomarker variables in patients with benign prostatic hyperplasia (BPH) without bladder diverticulum. The heatmap's color intensity and the ellipses' color indicate the strength and direction of the correlation, with red tones depicting positive correlations and blue tones indicating negative correlations. The size and orientation of the ellipses correspond to the strength and nature of the correlation; narrower ellipses represent stronger correlations, and wider ellipses represent weaker correlations. The numerical values within the heatmap's cells are the actual correlation coefficients (rho values), with 1.00 indicating a perfect positive correlation and -1.00 indicating a perfect negative correlation.

The implications of ROS, particularly 8-OHdG and 8-iso-prostaglandin F2α (8-Iso-PGF2α), have been explored in prostate cancer (PCa) onset and progression [36]. Elevated levels of these biomarkers in PCa patients, which normalize post-robot-assisted radical prostatectomy (RARP), underscore their potential utility in both clinical practice and future research. This normalization suggests the successful surgical removal of a key source of excess free radicals, indicating the radicality of the treatment and possibly predicting local recurrence. The findings suggest that measuring 8-OHdG and

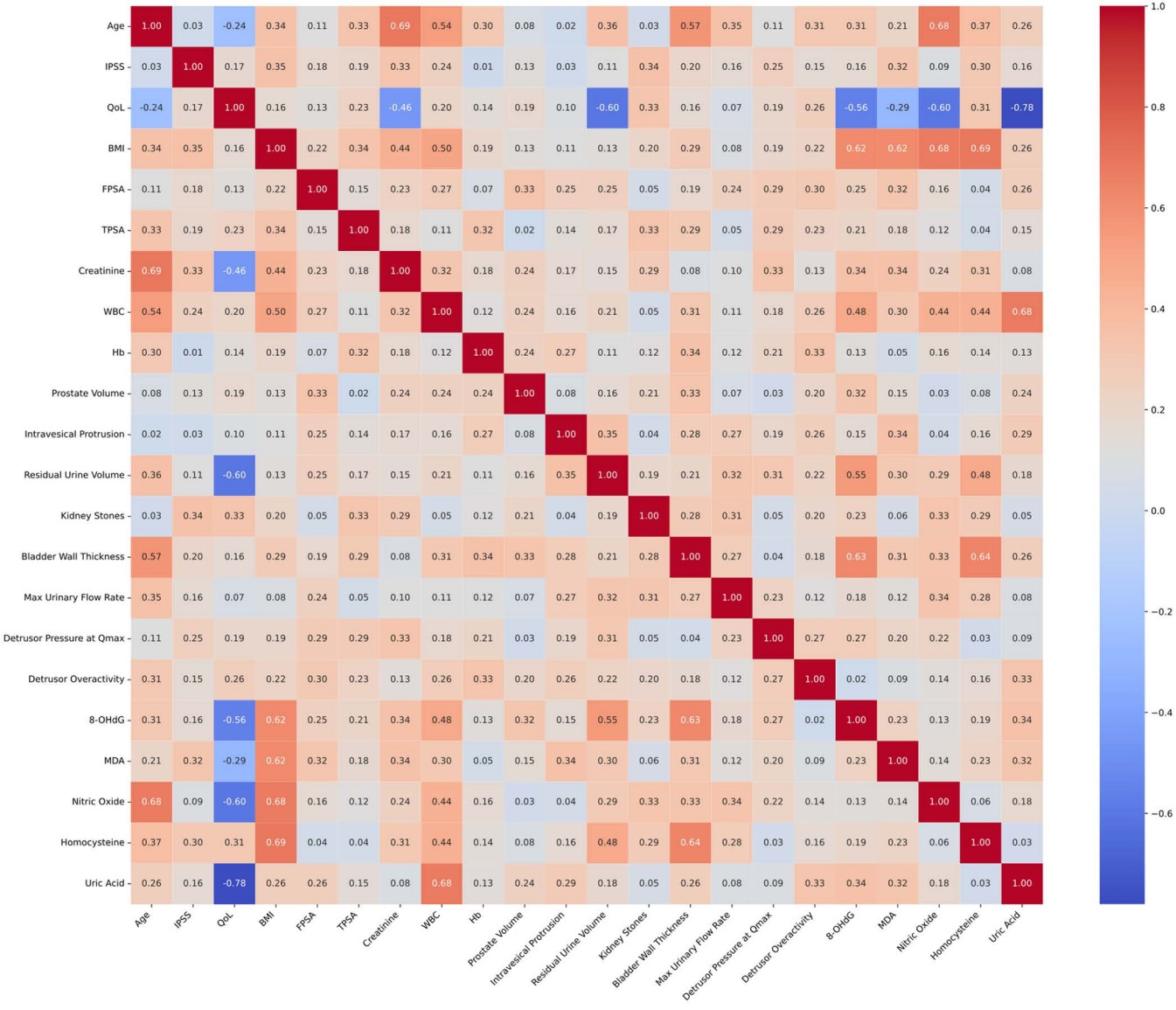

**Fig 5. Correlation matrix of clinical and oxidative stress biomarkers in BPH patients with bladder diverticulum.**

8-Iso-PGF2α levels could become an integral part of the pre- and post-operative care of PCa patients. Given their association with oxidative stress and inflammation, these biomarkers could help in refining patient risk stratification and monitoring post-surgical recovery. Furthermore, this study suggests the therapeutic potential of antioxidant therapies in prostate cancer (PCa) treatment. Antioxidant supplementation may help mitigate ROS-induced damage, potentially inhibiting PCa progression and complementing conventional therapies [37].

In designing this study, we aimed to explore a subset of BPH patients who present with bladder diverticula, a condition often overlooked in standard BPH research. Bladder diverticula, while not uncommon in severe BPH cases, represent a complication that can significantly alter the disease's progression and patient outcomes. By studying BPH patients with bladder diverticula, we address a literature gap and highlight therapeutic challenges.

We included patients with bladder diverticula to test whether their oxidative stress profiles and bladder dynamics differ from those without diverticula. This differentiation could offer critical insights into tailored therapeutic approaches that consider the unique pathophysiology of BPH in the context of concurrent bladder diverticula. Moreover, understanding these differences is essential for improving diagnostic accuracy and treatment efficacy in this more complicated clinical presentation.

Our findings reveal several unique insights into the interplay between BPH and bladder diverticula, which could have significant clinical implications. Firstly, the study demonstrated that patients with BPH and bladder diverticula exhibit elevated oxidative stress markers compared to those without diverticula. This suggests a heightened inflammatory response, potentially contributing to the exacerbation of urinary symptoms and increased risk of complications such as urinary tract infections and bladder stones.

Furthermore, the presence of bladder diverticula was associated with altered detrusor dynamics, as evidenced by variable Pdet values during the voiding phase. These findings underscore the mechanical challenges in managing bladder emptying in patients with both BPH and bladder diverticula, highlighting the need for specialized management strategies that address both the obstructive and diverticular components of the disease.

Additionally, our analysis indicates that patients with bladder diverticula tend to experience more severe symptoms and have a higher incidence of chronic bladder conditions, confirming the clinical significance of including these patients in BPH studies. By focusing on this cohort, our research contributes to a deeper understanding of BPH as a multifactorial disease that requires a multifaceted therapeutic approach.

In assessing the clinical implications of elevated Residual Urine Volume (RUV, post voiding) volumes observed in our study cohort, we performed a detailed analysis distinguishing between BPH and chronic urinary retention. Elevated RUV volumes are often associated with more severe forms of BPH but can also indicate underlying chronic urinary retention, which requires distinct clinical management strategies. In our study, RUV was measured using a bladder scanner post-void, with values exceeding 200 mL considered indicative of significant urinary retention based on current clinical guidelines [38].

The analysis revealed that a significant proportion of our cohort with high RUV also exhibited other symptoms indicative of severe obstruction, such as reduced flow rates and increased detrusor wall thickness. This suggests a severe obstruction likely due to enlarged prostate volume rather than functional urinary retention. These findings underscore the necessity of individualized therapeutic approaches that consider the severity of obstruction and RUV levels, potentially incorporating intermittent catheterization or surgical intervention for those with markedly elevated RUV volumes.

To ensure clarity in the study's focus on BPH and its differentiation from chronic urinary retention, our inclusion and exclusion criteria were stringently applied during the patient selection process. Patients eligible for inclusion were those diagnosed with BPH via clinical examination, including digital rectal examination and confirmed by ultrasound measurements of prostate volume. Importantly, to isolate the effects of BPH from other forms of urinary retention, we excluded individuals with a known history of chronic urinary retention that was not linked to BPH. Chronic urinary retention was defined as a condition where patients had a consistently high RUV of over 300 mL in multiple assessments without evidence of prostate enlargement or bladder outlet obstruction [39].

This exclusion was crucial in ensuring that the study population accurately represented BPH patients without confounding from other urological conditions that could independently affect bladder dynamics and RUV measurements. Such rigorous screening is vital for the accurate interpretation of RUV values in the context of BPH-related urinary retention and supports the reliability of our findings and conclusions drawn regarding the impact of prostate volume and bladder diverticula on urinary retention in BPH patients.

The correlation analysis revealed notable positive correlations between oxidative stress markers and specific clinical parameters in group A. 8-OHdG is a urine biomarker for oxidative stress [40]. A significant positive correlation was observed between the oxidative stress marker 8-OHdG and residual urine volume (rho = 0.68), indicating that higher levels

of oxidative stress are associated with increased residual urine. Similarly, nitric oxide showed a strong positive correlation with bladder wall thickness (rho = 0.70), suggesting that elevated nitric oxide levels might contribute to or result from changes in bladder wall morphology. Conversely, significant negative correlations were identified between nitric oxide (-0.76), creatinine (-0.70), MDA (-0.55), and uric acid (-0.52) with quality of life (QoL), highlighting those higher levels of these markers are associated with a decrease in patients' QoL, potentially due to the exacerbation of symptoms or overall disease burden. Some results were consistent with previous report that MDA [41] and uric acid [42] are negatively associated with QoL.

In group B, consisting of BPH patients with bladder diverticulum, the analysis underscored significant associations between oxidative stress markers and both BMI and bladder wall thickness. Nitric oxide (rho = 0.68) and homocysteine (rho = 0.69) were positively correlated with BMI, implying that oxidative stress may be linked with higher body mass, which are consistent with previous reports [43,44]. Furthermore, both nitric oxide (rho = 0.69) and homocysteine (rho = 0.64) showed strong positive correlations with bladder wall thickness, reinforcing the potential role of these markers in bladder pathology. Additionally, negative correlations with QoL were prominent, with markers such as 8-OHdG (-0.56), MDA (-0.29), nitric oxide (-0.60), and uric acid (-0.78) indicating that increased oxidative stress is associated with reduced QoL. These findings suggest a complex interplay between oxidative stress, clinical parameters, and patient well-being, emphasizing the need for targeted therapeutic strategies to mitigate oxidative stress and improve clinical outcomes.

Our study's findings on oxidative stress in BPH patients, with and without bladder diverticulum, both align with and build upon the insights from S1 and S2 Tables, giving us a deeper understanding of how bladder diverticulum ramps up oxidative stress in BPH. In our cohort, we found that group B (BPH with bladder diverticulum) had significantly higher levels of oxidative stress biomarkers, like 8-OHdG (1.93 ± 0.58 vs. 1.70 ± 0.73 ng/mg creatinine, p = 0.014) and MDA (2.46 ± 0.57 vs. 2.03 ± 0.57 µmol/L, p < 0.0001), compared to group A (BPH alone). This fits right in with what we saw in Table S1, where Ohtake et al. (2018) reported moderate (71.4%) to strong (28.6%) 8-OHdG expression in BPH tissues [26], Vital et al. (2016) tied elevated 8-OHdG to prostate weight [45], and Chang et al. (2018) and Kaya et al. (2017) found higher MDA levels in BPH patients versus controls [46,47]. S2 Table also backs this up, showing increased MDA and 8-OHdG in bladder cancer and other bladder issues (e.g., Ali-El-Dein et al., 2024; Jiang et al., 2023) [48,49], hinting at a common oxidative stress thread across bladder conditions. But what's unique about our work is how we've shown that bladder diverticulum pushes these markers even higher than in typical BPH, likely due to the much larger residual urine volume we measured (400.1 ± 252.0 vs. 150.7 ± 93.9 mL, p < 0.0001) and urinary stasis—a factor hinted at in S2 Table 's reports on detrusor overactivity and diverticulum (Jiang et al., 2023) but not pinned down with numbers like we did [48]. In group A, we noticed positive correlations like 8-OHdG with residual urine volume (rho = 0.68) and nitric oxide with bladder wall thickness (rho = 0.70), which line up with S1 Table's links between oxidative stress and structural changes [45, 50]. Meanwhile, in group B, we saw correlations with BMI (e.g., homocysteine, rho = 0.69) and bladder wall thickness (e.g., nitric oxide, rho = 0.69), tying into S2 Table's tissue remodeling observations [51] and S1 Table's high-fat diet BPH findings [52].

Our findings highlight the need for targeted antioxidant therapies, especially around bladder diverticulum and QoL. We found negative correlations between QoL and nitric oxide in group A (rho = -0.76) and uric acid in group B (rho = -0.78), which isn't something S1 and S2 Tables dig into much, since they rarely look at how oxidative stress hits patients' daily lives. S1 Table shows treatments like apocynin and silymarin cutting down oxidative stress in BPH [52,53], and S2 Table highlights similar wins with vinpocetine and caftaric acid in bladder conditions [54,55]. But our results suggest that for BPH patients with bladder diverticulum, we might need to step up our game with more targeted antioxidant therapies to tackle the extra oxidative load and its fallout—like higher creatinine (101.8 ± 27.6 vs. 56.1 ± 23.6 µmol/L, p < 0.0001) and inflammation (WBC counts, 7.0 ± 1.9 × 10⁹/L vs. 4.2 ± 1.3 × 10⁹/L, p < 0.0001). Unlike many reports in S2 Table that skip specific values, we've nailed down our biomarkers with solid numbers and statistical weight, setting a clear mark for others to build on. Plus, the way our group B data overlaps with S2 Table's bladder dysfunction trends (e.g., elevated 8-OHdG in diverticulum cases) makes us think there's a progression of oxidative stress from BPH to worse bladder issues, with diverticulum acting as a key player[48]. That's got us eager to dive into tailored treatments next.

## Clinical implications

Our findings highlight the significant role of oxidative stress in the pathophysiology of BPH, particularly in patients with bladder diverticulum. By demonstrating elevated levels of oxidative stress biomarkers such as 8-OHdG, MDA, nitric oxide metabolites, homocysteine, and uric acid in patients with bladder diverticulum, our study underscores the importance of considering oxidative stress as a therapeutic target. Clinicians can leverage these biomarkers to identify patients at higher risk of disease progression and complications. This can lead to more personalized treatment approaches, incorporating antioxidants or other therapies aimed at reducing oxidative stress, potentially improving patient outcomes and quality of life. For example, the correlation between oxidative stress markers and clinical parameters such as bladder wall thickness and residual urine volume provides actionable insights for managing patients more effectively by focusing on reducing oxidative damage and inflammation.

## Study limitations

The study presents several limitations that should be considered when interpreting its findings. The study primarily relies on observational data, and as such, it cannot establish causality between oxidative stress and the clinical manifestations observed in BPH patients with bladder diverticulum. The study does not appear to account for potential confounders or biases that could influence the relationship between oxidative stress biomarkers and BPH-related parameters, potentially affecting the accuracy of the findings. Additionally, the use of only ultrasound and CT scans may not capture all relevant clinical aspects of BPH and bladder diverticulum. Finally, the study does not provide long-term follow-up data, which is crucial for understanding the chronic nature of BPH and the potential long-term effects of oxidative stress in these patients. We did not differentiate between types of bladder diverticula (e.g., congenital vs. acquired), as our focus was on their presence and functional impact rather than their etiology. While we utilized transrectal ultrasound, CT scans, and cystometric measurements via a multichannel urodynamic system to identify and characterize diverticula—assessing parameters such as Pdet, bladder wall thickness, and urinary flow rates—this approach may overlook potential differences in pathophysiology or clinical outcomes associated with specific diverticulum types. Although we employed imaging segmentation to isolate diverticula and enhance the accuracy of detrusor function analysis, the lack of classification limits our ability to fully elucidate how diverticulum type might influence oxidative stress or bladder dysfunction in BPH patients, potentially restricting the generalizability of our findings to broader diverticulum-related contexts.

## The generalizability of this study

The inclusion of two BPH cohorts enhances generalizability, but the single-center design limits broader applicability. The use of both ultrasound imaging and CT scans enhances the reliability and validity of the findings related to prostate and bladder conditions. However, generalizability may be limited by the study's sample size, the specific population characteristics, and potential selection biases. Expanding future research to include diverse populations from multiple centers and various geographical locations would further validate these findings and improve their applicability to a broader patient demographic. Additionally, controlling for confounding variables and ensuring a more heterogeneous patient population can enhance the external validity of the results.

## Conclusions

In conclusion, this study highlights associations between oxidative stress and BPH with bladder diverticulum, suggesting a role in pathophysiology. The strong correlations between oxidative stress markers and clinical parameters in this subgroup underscore the potential for targeted interventions. Future studies should explore antioxidant therapies to manage oxidative stress in this population. This focus on a specific, clinically relevant BPH cohort allows for a more detailed examination of the disease's complexity and provides a foundation for future studies aimed at developing targeted interventions for

BPH patients with bladder diverticula. Our study not only adds to the body of knowledge on BPH but also highlights the importance of considering bladder diverticula in the clinical management of the disease, potentially leading to improved outcomes for this challenging patient population.

## Supporting information

**S1 Table. Oxidative stress in BPH.**
(DOCX)

**S2 Table. Oxidative stress in bladder diverticulum and cancer.**
(DOCX)

## Acknowledgments

We extend our heartfelt thanks to two anonymous reviewers whose strategic and insightful comments have significantly elevated the quality of our present manuscript. Their thoughtful feedback and expert suggestions helped us refine our arguments, strengthen our analysis of oxidative stress in BPH patients with and without bladder diverticulum, and improve the overall coherence and rigor of the study. By pointing out areas for enhancement and offering valuable perspectives, they've played a crucial role in sharpening our work, and we're deeply grateful for their time, expertise, and contributions to making this manuscript more robust and impactful. We express our gratitude to ChatGPT, Grok, and Google Grammar for their invaluable assistance in refining the language of our present manuscript. These tools have significantly enhanced the clarity, coherence, and grammatical accuracy of our writing, allowing us to effectively communicate our findings on oxidative stress in BPH patients with and without bladder diverticulum. ChatGPT and Grok, with their advanced language processing capabilities, provided insightful suggestions to improve our phrasing and structure, while Google Grammar helped us polish the text by catching subtle errors and ensuring consistency. Together, they've been instrumental in elevating the quality of this work, and we're thankful for the technological support that made this possible.

## Author contributions

**Conceptualization:** Shuijing Yin.

**Data curation:** Shuijing Yin.

**Formal analysis:** Yu Qiu.

**Investigation:** Shuijing Yin.

**Methodology:** Shuijing Yin, Yu Qiu.

**Project administration:** Yu Qiu.

**Resources:** Shuijing Yin.

**Software:** Shuijing Yin.

**Validation:** Yu Qiu.

**Writing – original draft:** Shuijing Yin.

**Writing – review & editing:** Yu Qiu.

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
