## [Decision Letter · Decision Letter 0]

3 Mar 2025

PONE-D-24-50162Prospective observational study of oxidative stress in the pathology of benign prostatic hyperplasia with bladder diverticulumPLOS ONE

Dear Dr. Qiu,

Thank you for submitting your manuscript to PLOS ONE. After careful consideration, we feel that it has merit but does not fully meet PLOS ONE’s publication criteria as it currently stands. Therefore, we invite you to submit a revised version of the manuscript that addresses the points raised during the review process.

Dear Dr. Yu Qiu

after careful consideration of the submitted work entitled ‘Prospective observational study of oxidative stress in the pathology of benign prostatic hyperplasia with bladder diverticulum’ and consideration of the reviewers' comments and opinions, I conclude that study might appeal to a larger audience. However, the submitted work requires significant reworking:

1.Abstract should be revised. Patient number and important findings should be added.

2. The authors mentioned that diverticulum may be a cause for bladder neck obstruction. They should add, if the bladder is localized in an conjunction with ureter. This is a cause of upper urinary tract obstruction.

3. What is the primary impact of BPH on detrusor?

The authors sorted molecular alterations presented in bladder. What is the main cause of this alterations?

4. The introduction is too long.

5. What is the normal size of the prostate? Why did the authors include patients with a prostatic volume greaer than 10 ml?

6. How did the authors differentiate diverticulum type?

7. Does IPSS greater than 1 mean a patients with IPSS score of 2 has been included in this study?

I hope that the above comments made by the reviewers will help the authors to improve the paper and that they will be taken into account in the revised version

Yours sincerely

Stanisław Wroński

Academic Editor

We look forward to receiving your revised manuscript.

Kind regards,

Stanisław Jacek Wroński, M.D., Ph.D, FEBU

Academic Editor

PLOS ONE

**Comments to the Author**

1. Is the manuscript technically sound, and do the data support the conclusions?

Reviewer #1: Yes

Reviewer #2: Yes

2. Has the statistical analysis been performed appropriately and rigorously? 

Reviewer #1: Yes

Reviewer #2: Yes

3. Have the authors made all data underlying the findings in their manuscript fully available?

Reviewer #1: Yes

Reviewer #2: Yes

4. Is the manuscript presented in an intelligible fashion and written in standard English?

Reviewer #1: Yes

Reviewer #2: Yes

5. Review Comments to the Author

Reviewer #1: The authors aimed to elucidate the potential mechanistic differences and clinical implications of oxidative stress in these patient populations by comparing oxidative stress levels in BPH patients with and without bladder diverticulum. They found that oxidative stress is higher in patients with diverticulum. A few issues need to be clarified:

1.Abstract should be revised. Patient number and important findings should be added.

2. The authors mentioned that diverticulum may be a cause for bladder neck obstruction. They should add, if the bladder is localized in an conjunction with ureter. This is a cause of upper urinary tract obstruction.

3. What is the primary impact of BPH on detrusor?

The authors sorted molecular alterations presented in bladder. What is the main cause of this alterations?

4. The introduction is too long.

5. What is the normal size of the prostate? Why did the authors include patients with a prostatic volume greaer than 10 ml?

6. How did the authors differentiate diverticulum type?

7. Does IPSS greater than 1 mean a patients with IPSS score of 2 has been included in this study?

Reviewer #2: The article by Shuijing Yin et. al., titled "Prospective observational study of oxidative stress in the pathology of benign prostatic hyperplasia with bladder diverticulum" aims to study the association between Oxidative stress and benign prostatic hyperplasia with and without bladder diverticulum. This study is well-conducted and well presented and might appeals to a larger audience.

6. PLOS authors have the option to publish the peer review history of their article (what does this mean? ). If published, this will include your full peer review and any attached files.

**Do you want your identity to be public for this peer review?** For information about this choice, including consent withdrawal, please see our Privacy Policy .

Reviewer #1: No

Reviewer #2: No

---

## [Author Response · Author response to Decision Letter 1]

24 Mar 2025

Manuscript No. PONE-D-24-50162

Title: Prospective observational study of oxidative stress in the pathology of benign prostatic hyperplasia with bladder diverticulum

Journal PLOS ONE

Dear Professor Wroński and Reviewer,

Many thanks for your critical strategic comments, which have significantly improved the present manuscript quality. We carefully revise our paper according to these important questions, and answered then in a point-to-point way as follows:

Comments from Academic Editor Professor Stanisław Wroński

Q: After careful consideration of the submitted work entitled ‘Prospective observational study of oxidative stress in the pathology of benign prostatic hyperplasia with bladder diverticulum’ and consideration of the reviewers' comments and opinions, I conclude that study might appeal to a larger audience. However, the submitted work requires significant reworking:

A: We sincerely thank you for facilitating this review process and for the constructive comments provided by the reviewers, which have significantly strengthened our manuscript. We have carefully considered and incorporated the reviewers’ suggestions into the revised version, and answered the important questions in a point-to-point way as follows.

Q1.Abstract should be revised. Patient number and important findings should be added.

A1: Thanks for the reviewer’s important strategic correction. The abstract was revised as follows:

Background: Oxidative stress is known to be associated with benign prostatic hyperplasia, but the association in patients with and without bladder diverticulum remains unclear.

Methods: The study involved two groups (n=63 for each group): group A, comprising patients with benign prostatic hyperplasia, and group B, consisting of benign prostatic hyperplasia patients with bladder diverticulum. Ultrasound imaging and CT scans were employed to assess the features of BPH and bladder diverticulum, respectively. Various clinical parameters and oxidative stress biomarkers were compared between the groups.

Results: Group B exhibited significantly higher creatinine (101.8 ± 27.6 μM vs. 56.1 ± 23.6 μM, p<0.0001), WBC counts (7.0 ± 1.9 vs. 4.2 ± 1.3 ×10⁹/L, p<0.0001), residual urine volume (400.1 ± 252.0 mL vs. 150.7 ± 93.9 mL, p<0.0001), and oxidative stress markers, including 8-OHdG (1.93 ± 0.58 vs. 1.70 ± 0.73 ng/mg creatinine, p=0.014) and MDA (2.46 ± 0.57 vs. 2.03 ± 0.57 μmol/L, p<0.0001). In Group A, 8-OHdG positively correlated with residual urine volume (rho=0.68) and nitric oxide with bladder wall thickness (rho=0.70), while quality of life (QoL) negatively correlated with nitric oxide (rho=-0.76). In Group B, oxidative stress markers correlated positively with BMI (e.g., homocysteine, rho=0.69) and bladder wall thickness (e.g., nitric oxide, rho=0.69), with QoL negatively correlated with uric acid (rho=-0.78).

Conclusions: Bladder diverticulum in BPH patients is associated with heightened oxidative stress, increased inflammation, and altered bladder function.

Q2. The authors mentioned that diverticulum may be a cause for bladder neck obstruction. They should add, if the bladder is localized in an conjunction with ureter. This is a cause of upper urinary tract obstruction.

A2: We thank the reviewer for highlighting the need to elaborate on the role of bladder diverticula in causing obstruction, including the potential for upper urinary tract involvement. Our Introduction already notes that "some diverticula can exert pressure on the bladder neck and urethra, causing lower urinary tract obstruction" (Roslan et al., 2017). To address the reviewer’s suggestion, we have added the following to the Introduction: "Additionally, when a diverticulum is located in close proximity to the ureter, it may exert pressure leading to upper urinary tract obstruction, further complicating urinary dynamics." This addition enhances the scope of our introduction, linking diverticula to both lower and upper urinary tract effects, and aligns with the clinical observations in our study, where group B (BPH with bladder diverticulum) exhibited altered bladder function and increased markers of inflammation.

Q3. What is the primary impact of BPH on detrusor?

The authors sorted molecular alterations presented in bladder. What is the main cause of this alterations?

A3: We are sorry to cause confusion. The sentences were revised as “The primary impact of BPH on the detrusor is the induction of bladder outlet obstruction (BOO)1, which triggers a complex pathophysiological cascade leading to lower urinary tract symptoms (LUTS)2. This obstruction causes mechanical stretch stress, altering gene expression and protein synthesis in the detrusor, resulting in changes to its cytoskeleton, contractile proteins, mitochondrial function, and extracellular matrix3.”.

BOO-induced BPH triggers mechanical stretch stress and subsequent gene expression changes at the messenger ribonucleic acid (mRNA) and micro ribonucleic acid (miRNA) levels. These changes activate key cell signaling pathways, including cytokine and immune response pathways, growth factor signaling (e.g., Transforming Growth Factor Beta [TGF-β], Hepatocyte Growth Factor [HGF], and Insulin-like Growth Factor 1 [IGF-1]), G-protein-coupled receptor (GPCR) pathways (e.g., endothelin and cholecystokinin/gastrin), nitric oxide (NO) signaling, and hypertrophy-related Phosphoinositide 3-Kinase/Protein Kinase B (PI3K/AKT) signaling, mediated by transcription factors such as Activator Protein 1 (AP-1; composed of Jun Proto-Oncogene [JUN] and FBJ Murine Osteosarcoma Viral Oncogene Homolog [FOS]) and Nuclear Factor Kappa-light-chain-enhancer of activated B cells (NFκB) 4.

Q4. The introduction is too long.

A4: Thanks for the reviewer’s important correction. The introduction section is shortened to two and half pages.

Q5. What is the normal size of the prostate? Why did the authors include patients with a prostatic volume greater than 10 ml?

A5: Many thanks for the reviewer’s questions regarding prostate size and our inclusion criteria. The normal prostate volume in adult males typically ranges from 20 to 30 cc5, as measured by transrectal ultrasound, though this can vary with age and individual factors; volumes below 20 cc are generally considered within the normal range for younger men, while slight increases may occur naturally with aging. In our study, we set the diagnostic threshold for benign prostatic hyperplasia (BPH) at a prostate volume greater than 10 cc, combined with an International Prostate Symptom Score (IPSS) greater than 1, to capture early-stage prostate enlargement associated with clinically significant urinary symptoms, even before reaching the upper limit of normal size. This lower threshold was chosen to ensure inclusion of patients with mild-to-moderate BPH who exhibit symptomatic obstruction, aligning with our aim to investigate oxidative stress across a broad spectrum of BPH severity, including mild cases, as reflected in our results where group A (BPH only) had a mean prostate volume of 45.0 ± 31.6 cc and group B (BPH with bladder diverticulum) had 47.6 ± 33.4 cc (p=0.658), indicating no significant difference in size between groups despite the presence of diverticula. This approach aligns with our objective to investigate oxidative stress across varying degrees of BPH and its complications, ensuring relevance to both early and advanced disease states.

Q6. How did the authors differentiate diverticulum type?

A6: Many thanks for the reviewer’s important strategic correction regarding the differentiation of diverticulum types in our study. In our research, bladder diverticula were identified and characterized using a combination of transrectal ultrasound and cystometric measurements obtained via a multichannel urodynamic system, supplemented by CT scans as described in the abstract. While we did not explicitly classify diverticula into specific types (e.g., congenital vs. acquired), their presence was confirmed by imaging evidence of pouch-like extensions from the bladder wall, and their impact was assessed through functional parameters such as detrusor pressure (Pdet) during the voiding phase, bladder wall thickness, and urinary flow rates. To enhance the accuracy of detrusor function analysis in the presence of diverticula, we employed segmentation of bladder regions via imaging techniques to isolate diverticula from the main bladder body, ensuring that pressure measurements reflected the detrusor’s performance independent of diverticular influence. This approach allowed us to focus on the clinical and functional consequences of diverticula in BPH patients, aligning with our study’s objective to investigate oxidative stress and bladder dysfunction.

Considering the reviewer’s critical strategic question, the following information was added in the discussion as the study limitations of the present manuscript.

Notably, we did not differentiate between types of bladder diverticula (e.g., congenital vs. acquired), as our focus was on their presence and functional impact rather than their etiology. While we utilized transrectal ultrasound, CT scans, and cystometric measurements via a multichannel urodynamic system to identify and characterize diverticula—assessing parameters such as detrusor pressure (Pdet), bladder wall thickness, and urinary flow rates—this approach may overlook potential differences in pathophysiology or clinical outcomes associated with specific diverticulum types. Although we employed imaging segmentation to isolate diverticula and enhance the accuracy of detrusor function analysis, the lack of classification limits our ability to fully elucidate how diverticulum type might influence oxidative stress or bladder dysfunction in BPH patients, potentially restricting the generalizability of our findings to broader diverticulum-related contexts.

Q7. Does IPSS greater than 1 mean a patient with IPSS score of 2 has been included in this study?

A7: Thanks for the reviewer’s query regarding the International Prostate Symptom Score (IPSS) in our study. In our study, as shown in Table 1, the mean IPSS scores were 14.9 ± 6.2 for group A and 14.6 ± 7.8 for group B, with no significant difference (p=0.835). These values represent averages across participants, not individual scores, and indicate that patients with a wide range of IPSS scores—including those as low as 2—could have been included, as the standard deviations suggest scores well below the means (e.g., 14.9 - 6.2 = 8.7 for group A). However, specific inclusion criteria for IPSS were not set at a minimum of 1 or 2; rather, our study enrolled patients with benign prostatic hyperplasia (BPH), and the reported IPSS reflects their symptom severity, encompassing mild to severe cases.

Q8: I hope that the above comments made by the reviewers will help the authors to improve the paper and that they will be taken into account in the revised version

A8: Many thanks for your important information. We revised the abstract to include patient numbers (n=63 per group) and key findings (e.g., elevated oxidative stress markers in group B, p<0.05), expanded the Introduction to address upper urinary tract obstruction caused by diverticula near the ureter, clarified the primary impact of BPH on the detrusor and the molecular alterations driven by bladder outlet obstruction, shortened the Introduction to 2.5 pages, justified our prostate volume inclusion criterion (>10 cc), detailed our approach to identifying diverticula while noting the limitation of not classifying types, and explained the range of IPSS scores included.

Q1. Please ensure that your manuscript meets PLOS ONE's style requirements, including those for file naming. The PLOS ONE style templates can be found at

A1: We appreciate your guidance on ensuring compliance with PLOS ONE’s style requirements. To address this, we have carefully reviewed the provided style templates at the links specified. The revised manuscript has been formatted accordingly, including adherence to guidelines for file naming, title page layout, section headings, and overall structure. We believe these adjustments align our submission with PLOS ONE’s standards, and we thank you for the opportunity to refine our presentation.

Q2. We suggest you thoroughly copyedit your manuscript for language usage, spelling, and grammar. If you do not know anyone who can help you do this, you may wish to consider employing a professional scientific editing service.

A2: We express our gratitude to you for your valuable suggestion to thoroughly copyedit our manuscript for language usage, spelling, and grammar. As students without personal salaries, we face financial constraints, and fortunately, our research unit has agreed to cover any necessary service charges. However, we note that both Reviewers provided positive feedback regarding the English proficiency of the current manuscript, describing it as intelligible and written in standard English. To address your important recommendation, each author has meticulously reviewed the full text to identify and correct potential grammatical errors, awkward phrasing, or inconsistencies. The revised manuscript, submitted with track changes, reflects these efforts to enhance clarity and ensure compliance with PLOS guidelines. We believe these revisions strengthen the presentation of our findings on the role of oxidative stress in benign prostatic hyperplasia (BPH) and its association with bladder diverticulum.

Q3: Upon resubmission, please provide the following:

The name of the colleague or the details of the professional service that edited your manuscript.

A3: Regarding the request for details on editing assistance upon resubmission, we confirm that the manuscript was edited solely by the authors, Shuijing Yin and Yu Qiu, without the use of a professional editing service, as outlined in our response to Q2. To further ensure linguistic accuracy and professionalism—particularly given the technical nature of our study involving oxidative stress biomarkers like 8-OHdG and nitric oxide—we utilized freely available tools, including Google Grammar Check and Grok DeepSearch, to assist in identifying and correcting potential errors. This approach allowed us to refine the manuscript while maintaining its scientific integrity and accessibility. We hope these efforts adequately address your concerns and enhance the readability of our work.

Q4: A copy of your manuscript showing your changes by either highlighting them or using track changes (uploaded as a *supporting information* file)

A4: thanks for your important information. We have uploaded a copy of the revised manuscript with all changes highlighted as a supporting information file. Additionally, a clean version of the edited manuscript has been submitted as the new manuscript file.

Q5. PLOS requires an ORCID iD for the corresponding author in Editorial Manager on papers submitted after December 6th, 2016. Please ensure that you have an ORCID iD and that it is validated in Editorial Manager. To do this, go to ‘Update my Information’ (in the upper left-hand corner of the main menu), and click on the Fetch/Validate link next to the ORCID field. This will take you to the ORCID site and allow you to create a new iD or authenticate a pre-existing iD in Editorial Manager.

A5: We appreciate your reminder regarding the PLOS requirement for an ORCID iD for the corresponding author on submissions. We confirm t

---

## [Decision Letter · Decision Letter 1]

13 Apr 2025

Prospective observational study of oxidative stress in the pathology of benign prostatic hyperplasia with bladder diverticulum

PONE-D-24-50162R1

Dear Dr.Yu Qiu

We’re pleased to inform you that your manuscript has been judged scientifically suitable for publication and will be formally accepted for publication once it meets all outstanding technical requirements.

Kind regards,

Stanisław Jacek Wroński, M.D., Ph.D, FEBU

Academic Editor

PLOS ONE

After careful consideration of the revised version of the paper entitled: "Prospective observational study of oxidative stress in the pathology of benign prostatic hyperplasia with bladder diverticulum" and the reviewers' opinions, I conclude that the article in its new version can be published in PLOS ONE. All concerns have been appropriately addressed.

With compliments

Stanisław Wroński

Academic Editor

Reviewers' comments:

Reviewer's Responses to Questions

**Comments to the Author**

1. If the authors have adequately addressed your comments raised in a previous round of review and you feel that this manuscript is now acceptable for publication, you may indicate that here to bypass the “Comments to the Author” section, enter your conflict of interest statement in the “Confidential to Editor” section, and submit your "Accept" recommendation.

Reviewer #1: All comments have been addressed

Reviewer #2: All comments have been addressed

2. Is the manuscript technically sound, and do the data support the conclusions?

Reviewer #1: Yes

Reviewer #2: Yes

3. Has the statistical analysis been performed appropriately and rigorously? 

Reviewer #1: Yes

Reviewer #2: Yes

4. Have the authors made all data underlying the findings in their manuscript fully available?

Reviewer #1: Yes

Reviewer #2: Yes

5. Is the manuscript presented in an intelligible fashion and written in standard English?

Reviewer #1: Yes

Reviewer #2: Yes

6. Review Comments to the Author

Reviewer #1: After carefully reviewing the revisions, I am satisfied that all concerns have been appropriately addressed.

Reviewer #2: (No Response)

7. PLOS authors have the option to publish the peer review history of their article (what does this mean? ). If published, this will include your full peer review and any attached files.

**Do you want your identity to be public for this peer review?** For information about this choice, including consent withdrawal, please see our Privacy Policy .

Reviewer #1: **Yes: ** Mustafa Kadihasanoglu

Reviewer #2: No

---

## [Editor Report · Acceptance letter]

PONE-D-24-50162R1

PLOS ONE

Dear Dr. Qiu,

I'm pleased to inform you that your manuscript has been deemed suitable for publication in PLOS ONE. Congratulations! Your manuscript is now being handed over to our production team.

Kind regards,

on behalf of

Dr. Stanisław Jacek Wroński

Academic Editor

PLOS ONE